# Directed Information $\gamma$-covering: An Information-Theoretic Framework for Context Engineering

## Abstract

We introduce **Directed Information $\gamma$-covering**, a simple but general framework for redundancy-aware context engineering. Directed information (DI), a causal analogue of mutual information, measures asymmetric predictiveness between chunks. If $\mathrm{DI}_{i \to j} \geq H(C_j) - \gamma$, then $C_i$ suffices to represent $C_j$ up to $\gamma$ bits. Building on this criterion, we formulate context selection as a $\gamma$-cover problem and propose a greedy algorithm with provable guarantees: it preserves query information within bounded slack, inherits $(1 + \ln n)$ and $(1 - 1/e)$ approximations from submodular set cover, and enforces a diversity margin. Importantly, building the $\gamma$-cover is *query-agnostic*: it incurs no online cost and can be computed once offline and amortized across all queries. Experiments on HotpotQA show that $\gamma$-covering consistently improves over BM25, a competitive baseline, and provides clear advantages in hard-decision regimes such as context compression and single-slot prompt selection. These results establish DI $\gamma$-covering as a principled, self-organizing backbone for modern LLM pipelines.

## 1 Introduction

Ever since the seminal work of Brown et al. (2020), which introduced GPT-3 and demonstrated the potential of few-shot prompting, *prompt engineering*—and later the broader field of *context engineering*—has flourished. Retrieval-augmented generation (RAG) (Lewis et al., 2020) and model–context protocols (MCP) (Hou et al., 2025) have made externalized systems the de facto way to handle long or dynamic context, appropriating techniques from information retrieval and vector databases. Yet despite the variety of approaches and framings, the core challenge remains unchanged: *How can we select, compress, and diversify context under strict budget constraints without losing essential information?*

While these advances have pushed the field forward, existing methods still rely heavily on sparse heuristics, ad hoc tricks, or query-dependent signals. This reliance highlights the need for a more principled foundation. Inspired by Heinz von Foerster's principles of *self-organization* (von Foerster, 1962) and Hermann Haken's *Synergetics* (Haken, 1977), we argue that **what is missing is a self-organizing principle** for context engineering. Like all other forms of information, context exhibits emergent patterns that can and should be leveraged, rather than imposed through manual heuristics. This perspective echoes parallel developments in self-supervised learning: LeCun (2022) explicitly argued that self-supervised learning is the central organizing principle for intelligence, a view further reinforced by the comprehensive survey of methods in Balestriero et al. (2023). We view our work as the natural counterpart of self-supervised learning in the domain of context engineering: just as self-supervised learning organizes *internal* representations, context engineering demands an information-theoretic, self-organizing mechanism for *external* knowledge.

We propose **Directed Information (DI)** as this principle. DI (Massey, 1990; Kim, 2008) describes a fine-grained, *asymmetric* predictive relationship among context chunks. Unlike symmetric measures such as entropy or perplexity, DI naturally induces directionality: one chunk may nearly determine another without the converse being true. This directional structure can be exploited to let context *self-organize*, guiding which chunks should be selected, compressed, or merged.

In this paper, we introduce **Directed Information $\gamma$-covering** as a self-organizing backbone for context engineering. Our approach provides a unified solution across diverse applications including reranking, context compression, and system prompt selection. Concretely, we make the following contributions:

1. **DI as a self-organizing principle.** We introduce directed information as a fine-grained, asymmetric measure that endows context with an emergent, self-organizing structure, in contrast to heuristic or query-dependent approaches.

2. **From query-dependent to query-agnostic.** We prove that DI bounds pointwise mutual information (PMI), bridging from expensive query-specific relevance signals to a query-agnostic framework. Crucially, this structure incurs no online cost: $\gamma$-covers can be computed once offline and amortized across all queries, enabling efficient large-scale deployment.

3. **Operationalizable via $\gamma$-covering.** We formulate context selection as a $\gamma$-cover problem and design a greedy algorithm that inherits classical set cover guarantees, making DI-based context engineering practical and efficient.

4. **Theoretical guarantees.** We establish bounds for soundness (information preservation), diversity (non-redundancy margins), and approximation ( $(1 + \ln n)$ and $(1 - 1/e)$ ), providing a principled foundation for selection, compression, and reranking.

5. **Empirical validation.** Across reranking, context compression, and system prompt selection on HotpotQA, we show consistent improvements over BM25 and clear advantages in hard-decision regimes (hard compression and minimum prompt selection), confirming the practical value of our framework.

Taken together, our results position **Directed Information $\gamma$-covering** as a self-organizing backbone for context engineering, bridging the gap between theoretical information measures and practical LLM retrieval pipelines.

## 2 RELATED WORK

**Context engineering**—the design and selection of input context—has emerged as a critical challenge in large language models (LLMs) (Mei et al., 2025). It subsumes many applications, ranging from retrieval-augmented generation (RAG) (Lewis et al., 2020) to reranking methods (Carbonell & Goldstein, 1998; Nogueira & Cho, 2019; Li et al., 2020). Among the most relevant lines of work to this paper are *information-theoretic approaches*. Peyrard (2019) introduced an entropy-based framework balancing redundancy, relevance, and informativeness for summarization. Khurana & Bhatnagar (2022) studied entropy as a signal in document summarization, while Li et al. (2023) proposed using self-information to compress context. Several recent works leverage token entropy to manage long contexts: Yao et al. (2024) designed SIRLLM, a system for long-term memory in infinite-length dialogues without fine-tuning, and Jung et al. (2024) used entropy with masked language models to distill a powerful summarizer. The LLMLingua family (Jiang et al., 2023; 2024) employs perplexity as a measure of information density for query-aware long-context compression; their later work (Pan et al., 2024) extends the approach to task-agnostic settings via a binary classification formulation. More recently, pointwise mutual information has been proposed as a gauge for RAG (Liu et al., 2025). Despite these advances, existing methods primarily rely on symmetric measures such as entropy or perplexity. To the best of our knowledge, there is no theoretical framework that leverages *directed information (DI)* (Massey, 1990; Kim, 2008). DI enables the definition of fine-grained, asymmetric predictive relations between context chunks, which we exploit to self-organize and compress context in a principled way.

**Information theory** has long provided a principled basis for selecting and compressing features. Rate–distortion theory (Shannon, 1959) formalizes the idea of retaining information up to a fidelity tolerance. The information bottleneck method (Tishby et al., 1999; Strouse & Schwab, 2017) extends this perspective, seeking minimal sufficient representations that preserve predictive power for a target variable. Related ideas appear in approximate sufficiency and generalization analysis in learning theory (Xu & Raginsky, 2017). In natural language processing, mutual information (MI) has been widely used for feature selection and retrieval scoring (e.g., Liu et al. (2025)).

**The set cover problem** is a classical NP-hard combinatorial problem (Karp, 1972), where the greedy algorithm achieves a $(1 + \ln n)$-approximation (Johnson, 1974). For the budgeted maximum coverage variant, greedy achieves a $(1 - 1/e)$ approximation (Nemhauser et al., 1978). These guarantees rely on the monotonicity and submodularity of the coverage objective. Extensions of these ideas appear in influence maximization (Kempe et al., 2003), where greedy selection of seed nodes approximates the spread of influence under diffusion models.

## 3 DIRECTED INFORMATION $\gamma$-COVERING

### 3.1 POINTWISE MUTUAL INFORMATION AND DIRECTED INFORMATION

First, we formally introduce pointwise mutual information (PMI) and directed information (DI), and establish that PMI can be bounded by DI. This result serves as the cornerstone of our framework, providing the bridge from query-dependent to query-agnostic measures.

Let $q$ denote a query and $\{C_i\}$ a collection of candidate chunks (token sequences). Let $p^\star$ be the reference distribution (the "ideal" language model).

**Definition 3.1** (Task-conditioned PMI). For any query $q$ and chunk $C$, define

$$\mathrm{PMI}^\star(q; C) \;=\; \log \frac{p^\star(q \mid C)}{p^\star(q)} \;=\; I^\star(q; C).$$

Intuitively, the greater the mutual information shared between a context chunk and a query, the more effectively the context can contribute to answering the query. Since PMI is query-dependent, it introduces substantial computational overhead at runtime. A more desirable alternative is a query-agnostic structure that can bound PMI. To this end, we propose leveraging Directed Information (DI) (Massey, 1990) between context chunks to facilitate context selection.

**Definition 3.2** (Directed Information (Massey, 1990)). For token sequence $C_j = (y_{j,1}, \ldots, y_{j,T_j})$, the *directed information* from $C_i$ to $C_j$ is

$$\mathrm{DI}_{i \to j} \;=\; \sum_{t=1}^{T_j} I^\star\big(C_i^{\leq t}; y_{j,t} \mid y_{j,<t}\big).$$

DI is a causal analogue of MI and can be used to bound both MI and PMI (proof in A.1).

**Lemma 3.1** (PMI coupling bounds). For any $q, C_i, C_j$ under $p^\star$,

$$\mathrm{PMI}^\star(q; C_i) - H^\star(C_i \mid C_j) \;\leq\; \mathrm{PMI}^\star(q; C_j) \;\leq\; \mathrm{PMI}^\star(q; C_i) + H^\star(C_j \mid C_i).$$

By Massey's decomposition (Massey, 1990; Kim, 2008), $I(C_i; C_j) = \mathrm{DI}_{i \to j} + \mathrm{DI}_{j \to i}$, and $H(C_j | C_i) = H(C_j) - I(C_i; C_j)$, we immediately have $H(C_j | C_i) \leq H(C_j) - \mathrm{DI}_{i \to j}$, and symmetrically $H(C_i | C_j) \leq H(C_i) - \mathrm{DI}_{j \to i}$. Hence the following corollaries.

**Corollary 3.1.1** (Pruning rule). If $\mathrm{PMI}^\star(q; C_i) \leq \tau$ and $\mathrm{DI}_{i \to j} \geq H^\star(C_j) - \gamma$, then

$$\mathrm{PMI}^\star(q; C_j) \;\leq\; \tau + \gamma.$$

**Corollary 3.1.2** (Promotion rule). If $\mathrm{PMI}^\star(q; C_j) \geq \tau$ and $\mathrm{DI}_{j \to i} \geq H^\star(C_i) - \gamma$, then

$$\mathrm{PMI}^\star(q; C_i) \;\geq\; \tau - \gamma.$$

The pruning and promotion corollaries are the core rules of our theoretical framework. Intuitively, if chunk $C_i$ strongly predicts $C_j$, then including $C_j$ becomes unnecessary if $C_i$ is already selected. Conversely, if $C_j$ is in the context, replacing it with $C_i$ should yield comparable, if not superior, performance.

### 3.2 EMPIRICAL PREDICTIVENESS

While the directed information $\mathrm{DI}_{i \to j}$ provides the theoretically correct measure of directional predictiveness, computing it exactly is not practical: it requires expectations under the true distribution $p^\star$ and summing over all possible token prefixes. In practice, we approximate $\mathrm{DI}_{i \to j}$ by an empirical NLL-drop estimator $\hat{w}_{i \to j}$.

**Definition 3.3** (Empirical predictiveness). Given a parametric LM $p_\theta$, define the *empirical predictiveness score*

$$\hat{w}_{i \to j} = \frac{1}{T_j} \Big( \text{NLL}_\theta(C_j) - \text{NLL}_\theta(C_j \mid C_i) \Big),$$

where $\text{NLL}_\theta(C_j) = -\sum_{t=1}^{T_j} \log p_\theta(y_{j,t} \mid y_{j,<t})$ denotes the negative log-likelihood.

Next we show the empirical NLL-drop estimator $\hat{w}_{i \to j}$ is a consistent proxy under mild assumptions.

**Theorem 3.2** (Estimator consistency). Assume (A1) bounded log-likelihood error $\sup_z |\log p_\theta(z) - \log p^\star(z)| \le \epsilon$, and (A2) per-token losses are sub-Gaussian. Then

$$\Big| \hat{w}_{i \to j} - \frac{1}{T_j} \sum_{t=1}^{T_j} I^\star(C_i^{\le t}; y_{j,t} \mid y_{j,<t}) \Big| \le \epsilon + O_\mathbb{P}\Big( \frac{1}{\sqrt{T_j}} \Big).$$

*Sketch.* (See A.2 for full proof) Compare the two conditional log-likelihoods tokenwise; apply Assumption A1 for approximation error and Hoeffding–Azuma for concentration of averages. □

Theorem 3.2 justifies replacing the ideal directed information $\text{DI}_{i \to j}$ with its empirical counterpart $\hat{w}_{i \to j}$. Building on this result, Theorem 3.3 establishes that, with high probability, our estimator yields valid pruning guarantees (proof in A.3).

**Theorem 3.3** (Safe pruning under estimation error). Let $\delta_i, \delta_j, \eta_j, \epsilon_{ij} \ge 0$ be numbers such that with probability at least $1 - \alpha$ the following hold simultaneously:

$$\big| \widehat{\text{PMI}}(q; C_i) - \text{PMI}^\star(q; C_i) \big| \le \delta_i,$$
$$\big| \widehat{\text{PMI}}(q; C_j) - \text{PMI}^\star(q; C_j) \big| \le \delta_j,$$
$$\big| \hat{H}(C_j) - H^\star(C_j) \big| \le \eta_j,$$
$$\big| \hat{w}_{i \to j} - \text{DI}_{i \to j} \big| \le \epsilon_{ij}.$$

Then, with probability at least $1 - \alpha$,

$$\widehat{\text{PMI}}(q; C_j) \le \widehat{\text{PMI}}(q; C_i) + \hat{H}(C_j) - \hat{w}_{i \to j} + (\delta_i + \delta_j + \eta_j + \epsilon_{ij}).$$

With these preliminaries in place, we now introduce the $\gamma$-covering algorithm.

### 3.3 GREEDY $\gamma$-COVERING ALGORITHM

First, we formally define $\gamma$-**covering** and $\gamma$-**coverage set**.

**Definition 3.4** ($\gamma$-covering edge). For two context chunks $C_i$ and $C_j$, we say that $i$ $\gamma$-covers $j$ if

$$\text{DI}_{i \to j} \ge H(C_j) - \gamma$$

Our $\gamma$-cover criterion follows the spirit of rate–distortion theory (Shannon, 1959) and information bottleneck methods (Tishby et al., 1999), where sufficiency is relaxed up to a fidelity tolerance. Here, $\gamma$ quantifies a tolerance in bits: we require that the residual uncertainty $H(C_j|C_i)$ be at most $\gamma$. Similar "$\epsilon$-sufficient" definitions appear in feature selection and approximate Markov sufficiency (e.g., (Xu & Raginsky, 2017)).

Intuitively, $i$ $\gamma$-covering $j$ means that directed information from $C_i$ to $C_j$ nearly saturates the entropy of $C_j$, leaving at most $\gamma$ bits unexplained. In other words, $C_i$ almost fully predicts $C_j$.

**Definition 3.5** ($\gamma$-coverage set). Given a chunk $C_i$, its $\gamma$-coverage set is

$$\text{Cov}_\gamma(i) = \{ j \mid \text{DI}_{i \to j} \ge H(C_j) - \gamma \}$$

**Remark**. Because DI is directional, it may hold that $i$ $\gamma$-covers $j$ but not vice versa. This asymmetry highlights the predictive directionality and is central to our framework.

**Input:** Candidate chunks $\{C_i \mid i \in [1, M]\}$; coverage sets $\text{Cov}_\gamma(i)$; budget $k$.
**Output:** Representative set $S$.
Initialize $S \leftarrow \varnothing, U \leftarrow [1, M]$ (uncovered items);
**while** $|S| < k$ *and* $U \neq \varnothing$ **do**
  pick $i^\star = \arg\max_{i \notin S} |\text{Cov}_\gamma(i) \cap U|$;
  $S \leftarrow S \cup i^\star$;
  $U \leftarrow U \setminus \text{Cov}_\gamma(i^\star)$;
**end**
**return** $S$

**Algorithm 1:** Greedy $\gamma$-covering

We now introduce the **Greedy $\gamma$-covering algorithm**, which operationalizes the $\gamma$-covering definitions into a practical selection procedure.

Given candidate chunks $\{C_i\}$ and their $\gamma$-coverage sets $\text{Cov}_\gamma(i)$, algorithm 1 selects a representative subset $S \subseteq [1, M]$. At each step, the algorithm adds the chunk that covers the largest number of currently uncovered items. The process repeats until either all items are covered or a budget constraint is met.

The set cover objective $f(S) = |\cup_{i \in S}\text{Cov}_\gamma(i)|$ is monotone and submodular (Nemhauser et al., 1978), and the problem of finding a minimum cover is NP-hard (Karp, 1972). Greedy achieves a $(1+\ln n)$-approximation for unconstrained set cover (Johnson, 1974) and a $(1-1/e)$-approximation for the budgeted/max-coverage variant (Nemhauser et al., 1978), as formally stated in theorem 3.4

**Theorem 3.4** (Approximation guarantees of Greedy $\gamma$-covering). Let $f(S) = \big| \cup_{i \in S} \text{Cov}_\gamma(i) \big|$ denote the $\gamma$-cover objective. Then Algorithm 1 achieves a $(1 + \ln n)$-approximation in the unconstrained setting. Under a budget of $k$ representatives, the algorithm guarantees a $(1 - 1/e)$-approximation.

In addition, Algorithm 1 admits a simplified "static" variant, obtained by removing the update step $U \leftarrow U \setminus \text{Cov}_\gamma(i^\star)$. In this case, items are ranked once according to their singleton coverage $|\text{Cov}_\gamma(i)|$, resulting in significantly lower computational cost. However, compared to the "dynamic" version, the "static" algorithm achieves only a $\frac{1}{k}$-approximation in the worst case (Khuller et al., 1999), as formalized below. The "static" variant finds application in the diffusion algorithm in section 5.1

**Proposition 3.1** (Static greedy). Selecting the $k$ items with the largest singleton coverage yields at best a $\frac{1}{k}$-approximation to the optimal budgeted solution in the worst case.

Algorithm 1 also admits a **clustering interpretation**: each selected representative $C_i$ defines a cluster containing all items $C_j$ it $\gamma$-covers. The greedy procedure can thus be viewed as a merge process: iteratively add the most influential node, merge its cluster, and continue until the budget is exhausted.

### 3.4 SOUNDNESS OF $\gamma$-REPRESENTATIVES

We slightly abuse notation by allowing $i$ to denote a node or the cluster rooted at $i$.

**Theorem 3.5** (Soundness of $\gamma$-cover representatives). Let $U$ be a set of candidate chunks and let $S \subseteq U$ be a set of representatives such that for every $j \in U \setminus S$ there exists $i \in S$ with $DI_{i \to j} \geq H(C_j) - \gamma$ (i.e., $i$ $\gamma$-covers $j$). Suppose the estimation slacks $(\delta_i, \delta_j, \eta_j, \epsilon_{ij})$ hold with probability at least $1 - \alpha$ as in Theorem 3.3. Then, with probability at least $1 - \alpha$,

$$I(q; U) \leq I(q; S) + \sum_{j \in U \setminus S} [\gamma + \delta_i + \delta_j + \eta_j + \epsilon_{ij}]$$

in particular, if all slacks are bounded by $\bar{\delta}$,

$$I(q; U) \leq I(q; S) + |U \setminus S|(\gamma + 3\bar{\delta})$$

Proof is in A.4. Theorem 3.5 formalizes that, **once a set $S$ $\gamma$-covers the rest, keeping only the root of $S$ preserves query information up to s small additive tolerance**. The PMI-DI coupling and the empricial "safe pruning" inequality are the only ingredients.

### 3.5 Diversity Margin Among Representatives

Another useful property of the $\gamma$-coverage set is that it provides a lower bound on the diversity of the selected cluster roots. Diversity, in turn, is a valuable attribute in context engineering (Lewis et al., 2020; Zamani & Croft, 2018; Carbonell & Goldstein, 1998), as it promotes complementary information and reduces redundancy.

**Proposition 3.2** (Diversity margin)**.** Let $S$ be any set of representatives produced by a $\gamma$-cover (i.e., no $i \in S$ $\gamma$-covers any other $j \in S$). Then

$$\min_{i \neq j \in S} \min\{H(C_i \mid C_j), H(C_j \mid C_i)\} > \gamma.$$

Equivalently, using Massey's decomposition $I(C_i; C_j) = DI_{i \to j} + DI_{j \to i}$,

$$\min_{i \neq j \in S} \{H(C_i) - \mathrm{DI}_{j \to i}, H(C_j) - \mathrm{DI}_{i \to j}\} > \gamma.$$

Proof is in A.5. Proposition 3.2 provides an information-theoretic margin: any two chosen representatives differ by at least $\gamma$ bits of irreducible uncertainty in at least one direction. This formalizes the intuitive "non-redundancy" (diversity) benefit of $\gamma$-covering and complements the approximation guarantees in Theorem 3.4.

## 4 Context Compression and System Prompt Selection

This section applies the $\gamma$-covering algorithm to two practical settings: *context compression* and *system prompt selection*. Our goal is to evaluate whether the theoretical guarantees of $\gamma$-covering—soundness and diversity—translate into empirical gains under controlled conditions.

We design experiments to carefully control sensitive hyperparameters, specifically the number of context chunks before and after compression. To create an idealized evaluation condition, we construct test inputs where the "optimal" number of relevant chunks is known. Concretely, we use the `distractor` subset of HotpotQA (Yang et al., 2018), which provides both a set of gold supporting facts (`supporting_facts`) and several distractor chunks. Let $s$ denote the number of gold supporting facts and $d$ the number of distractors. For each example, we retrieve all $s + d$ chunks (the full gold + distractor set), and then compress to the following sizes:

$$s + d - 1, \ \ s + d - 2, \ \ s, \ \ s - 1, \ \ s - 2.$$

Note that the cases $s - 1$ and $s - 2$ constitute *hard compression*, since at least one gold supporting fact is squeezed out. In contrast, all other cases are *soft compression*, where all gold facts can in principle be retained.

We compare $\gamma$-covering against a query-dependent PMI baseline, which ranks chunks by $\mathrm{PMI}(q; C)$ and keeps the top $k$. For each setting, we run 5 trials over randomly sampled 2,000 HotpotQA examples, reporting mean $\pm$ standard deviation for exact match (EM) and F1. To assess significance, we compute paired single-tailed $t$-tests against the PMI baseline.

**Results for context compression** is summarized in Table 1. We observe that in **hard compression** $(s - 1, s - 2)$, $\gamma$-covering achieves significantly higher EM and F1 than PMI (with $p < 0.05$). Here, the algorithm must discard some gold facts; $\gamma$-covering makes these decisions by favoring chunks with maximal predictive coverage, which aligns with our theoretical soundness guarantee. By contrast, PMI is unable to distinguish which gold chunk can safely be removed without severe information loss.

Although PMI performs better in soft compression in our limited experiments, we note that HotpotQA may not be a representative dataset for this setting, as it contains distractors, which is query-dependent, but relatively little redundancy. Moreover, PMI is query-dependent and incurs substantial online computational cost, whereas $\gamma$-covering operates offline and its cost can be amortized.

Table 1: Prompt compression results on HotpotQA. **Compressed** $K$ ($C_K$) denotes the number of context chunks remaining after compression.

| $C_K$ | EM (%) ↑ | | | F1 (%) ↑ | | |
|---|---|---|---|---|---|---|
| | **PMI** | $\gamma$-**cover** | $p$-value ↓ | **PMI** | $\gamma$-**cover** | $p$-value ↓ |
| $s-2$ | $31.33 \pm 0.69$ | $33.43 \pm 1.31$ | $2.91e-3$ | $36.49 \pm 0.60$ | $38.51 \pm 1.00$ | $4.69e-3$ |
| $s-1$ | $35.09 \pm 0.64$ | $36.57 \pm 1.23$ | $9.14e-3$ | $40.23 \pm 0.51$ | $41.83 \pm 0.83$ | $2.50e-3$ |
| $s$ | $46.69 \pm 0.70$ | $43.71 \pm 1.38$ | $4.93e-4$ | $52.00 \pm 0.40$ | $49.01 \pm 1.14$ | $6.52e-4$ |
| $s+d-2$ | $52.59 \pm 0.70$ | $49.60 \pm 0.59$ | $8.67e-4$ | $57.33 \pm 0.55$ | $54.10 \pm 0.27$ | $2.68e-4$ |
| $s+d-1$ | $56.20 \pm 0.75$ | $53.24 \pm 1.09$ | $6.89e-4$ | $60.61 \pm 0.70$ | $57.81 \pm 0.57$ | $8.57e-4$ |

**For system prompt selection**, we use the same experimental setup to evaluate *system prompt selection*, this time varying the retained budget $C_K \in \{1, 2\}$. This forces the model to keep only the most critical instructions or exemplars. The results (table 2) mirror the compression study. When $C_K = 1$, $\gamma$-covering significantly outperforms PMI. With a single slot available, the query-agnostic predictive criterion of $\gamma$-covering is better at identifying a representative prompt that subsumes the information of others. When $C_K = 2$, PMI performs better. Again, the average number of chunks in the ground truth is approximately 2.4. Hence, $C_K = 2$ can be regarded as partially within the soft compression regime.

Table 2: System prompt selection results on HotpotQA. **Compressed** $K$ ($C_K$) denotes the number of context chunks in the system prompt.

| $C_K$ | EM (%) ↑ | | | F1 (%) ↑ | | |
|---|---|---|---|---|---|---|
| | **PMI** | $\gamma$-**cover** | $p$-value ↓ | **PMI** | $\gamma$-**cover** | $p$-value ↓ |
| 1 | $30.38 \pm 0.86$ | $33.11 \pm 1.32$ | $7.42e-4$ | $35.46 \pm 0.66$ | $37.99 \pm 1.12$ | $2.14e-3$ |
| 2 | $43.78 \pm 0.36$ | $41.10 \pm 1.21$ | $2.21e-3$ | $49.08 \pm 0.27$ | $46.42 \pm 1.00$ | $2.19e-3$ |

## 5 RERANKING

Reranking is a standard post-processing step in retrieval pipelines: a base retriever produces an initial candidate set, which is then reordered by a stronger or more specialized model (Nogueira & Cho, 2019; Li et al., 2020; Carbonell & Goldstein, 1998; Lewis et al., 2020). While the $\gamma$-covering algorithm (Algorithm 1) naturally induces an ordering of context chunks—which we call the $\gamma$-*covering selection order*—applying this order directly in reranking does not always yield satisfying results. The reason is that retrievers already provide query-dependent scores indicating relevance to the input, whereas the $\gamma$-covering order is entirely query-agnostic. To combine their strengths, we adopt a *Lagrangian fusion* approach (Cao et al., 2007; Metzler & Croft, 2007), interpolating between the retriever's relevance scores and the structural ordering induced by the $\gamma$-covering graph.

### 5.1 DIFFUSION ALGORITHM

Algorithm 2 presents our **graph-aware reranker**, DIG-R (*Directional Information Graph for Reranking*). DIG-R is designed as a plug-and-play component that can be integrated into any retrieval system. Given a query and retriever scores, DIG-R refines the scores by diffusing relevance over the directional information graph. The algorithm implements a single-step diffusion, though multi-step diffusion is natural. Crucially, the computation of edge weights $\hat{w}_{i \rightarrow j}$ can be performed offline and amortized across queries, ensuring scalability. See A.6 for formal complexity and termination guarantee of Algorithm 2.

**Input:** Query $q$; retriever scores $r^{(0)}$ over neighborhood $\mathcal{N}$; damping $\alpha \in (0, 1)$.
**Output:** Top-$k$ chunks under token budget $B$.
**foreach** $i, j \in \mathcal{N}$ **do**
$\quad |\quad$ compute $\hat{w}_{i \to j} = \text{NLL}(C_j) - \text{NLL}(C_j \mid C_i)$
**end**
normalize columns of $W = [\hat{w}_{i \to j}]$ to form transition matrix $P$ ;
$r^{(1)} \leftarrow \alpha r^{(0)} + (1 - \alpha) P^\top r^{(0)}$ ;
return top-$k$ chunks by $r^{(1)}$ whose combined length $\leq B$ ;

$\qquad$ **Algorithm 2:** DIG-R: Graph-aware Reranking via Directional Information

## 5.2 EXPERIMENTS OF DIG-R

We evaluate our proposed reranker on top of a strong BM25 retriever. We use `pyserini` (Lin et al., 2021) with its prebuilt BM25 index (Robertson & Zaragoza, 2009) over HotpotQA (Yang et al., 2018), specifically the `beir-v1.0.0-hotpotqa.flat` index. Llama3.2-3B is used as the reader. We report both exact match (EM) and token-level F1, following standard HotpotQA evaluation. We adopt the same protocol to run each configuration is run with five independent trials over 2,000 randomly sampled examples.

Table 3: Reranking results on HotpotQA. **Retriever** $K$ ($RT_K$) is the number of context chunks returned by the retriever. **Reader** $K$ ($RD_K$) is the number of chunks passed to the reader. We tune $\alpha = 0.88$ once and fix it across all settings. Crossed-out $p$-values indicate non-significance at the 95% confidence level.

| $RT_K, RD_K$ | EM (%) ↑ | | | F1 (%) ↑ | | |
| | **BM25** | **DIG-R** | $p$-value ↓ | **BM25** | **DIG-R** | $p$-value ↓ |
|---|---|---|---|---|---|---|
| 4, 4 | $30.79 \pm 1.31$ | $31.23 \pm 1.13$ | $4.05e-3$ | $32.71 \pm 1.04$ | $32.90 \pm 0.96$ | $5.29e-3$ |
| 8, 4 | $30.79 \pm 1.31$ | $30.99 \pm 1.26$ | ~~0.0771~~ | $32.71 \pm 1.04$ | $32.90 \pm 1.14$ | $0.0308$ |
| 16, 8 | $30.63 \pm 1.26$ | $31.08 \pm 1.26$ | ~~0.0556~~ | $31.80 \pm 1.04$ | $31.88 \pm 0.83$ | ~~0.358~~ |

As shown in Table 3, DIG-R yields modest but consistent gains in both EM and F1 across configurations. Improvements are statistically significant when the reader consumes exactly the retrieved set ($RT_K = 4, RD_K = 4$), where reranking directly reshapes the context order. These results echo the observation of Liu et al. (2025) that context ordering often makes subtle differences: fusing the retriever's query-dependent ranking with the $\gamma$-covering selection order appears to produce contexts that are more LLM-friendly.

## 6 ABLATION STUDY

**First we present ablation study on MI vs. DI.** Although our theoretical framework is developed using DI, a similar formulation can be obtained by replacing DI with mutual information (MI). Since $I(C_i; C_j) = \text{DI}_{i \to j} + \text{DI}_{j \to i}$, one can derive analogous bounds, albeit slightly looser than those based on DI. We repeated several experiments with DI replaced by MI and observed a statistically significant degradation in performance (see Table 4).

Table 4: Replacing DI with MI. **Compressed** $K$ ($C_K$) denotes the number of context chunks remaining after compression. Crossed-out $p$-values indicate non-significance at the 95% confidence level.

| $C_K$ | EM (%) ↑ | | | F1 (%) ↑ | | |
| | **DI** | **MI** | $p$-value ↓ | **DI** | **MI** | $p$-value ↓ |
|---|---|---|---|---|---|---|
| $s-2$ | $33.43 \pm 1.31$ | $32.71 \pm 1.13$ | $0.108$ | $38.51 \pm 1.00$ | $37.84 \pm 0.93$ | $5.22e-3$ |
| 1 | $33.11 \pm 1.32$ | $32.08 \pm 1.10$ | $1.90e-3$ | $37.99 \pm 1.12$ | $37.16 \pm 0.95$ | $9.68e-4$ |

**We then ablate the dynamic $\gamma$-covering by replacing it with its static variant.** The *static* variant of the $\gamma$-covering algorithm ranks items once by their singleton coverage size and selects the top $k$,

without recomputing marginal gains. This makes it computationally attractive for reranking, since the resulting static order can be easily fused with retriever scores or other ranking signals.

Theoretically, static selection can be much weaker: it admits only a $\frac{1}{k}$-approximation in the worst case (Khuller et al., 1999), compared to the $(1 - 1/e)$ bound of dynamic greedy (Nemhauser et al., 1978). Nevertheless, our empirical results suggest that such worst-case behavior rarely materializes in practice. As shown in Table 5, the static variant exhibits only a very slight degradation compared to the dynamic version, while being significantly simpler and easier to integrate into reranking pipelines.

Table 5: Replacing dynamic clustering with static clustering. **Compressed** $K$ ($C_K$) denotes the number of context chunks remaining after compression. Crossed-out $p$-values indicate non-significance at the 95% confidence level.

| $C_K$ | EM (%) ↑ | | | F1 (%) ↑ | | |
|---|---|---|---|---|---|---|
| | **Dynamic** | **Static** | $p$-value ↓ | **Dynamic** | **Static** | $p$-value ↓ |
| $s - 2$ | $33.43 \pm 1.31$ | $33.39 \pm 1.37$ | ~~0.307~~ | $38.51 \pm 1.00$ | $38.54 \pm 0.99$ | ~~0.250~~ |
| $1$ | $33.11 \pm 1.32$ | $33.08 \pm 1.35$ | ~~0.187~~ | $37.99 \pm 1.12$ | $38.04 \pm 1.08$ | ~~0.136~~ |

**Finally, we ablate DIG-R.** Table 6 compares reranking results using $\gamma$-order with those of the DIG-R algorithm. Accuracy decreases slightly, though the difference is not statistically significant.

Table 6: Replacing DIG-R with $\gamma$-order. **Retriever** $K$ ($RT_K$) is the number of context chunks returned by the retriever. **Reader** $K$ ($RD_K$) is the number of chunks passed to the reader. Crossed-out $p$-values indicate non-significance at the 95% confidence level.

| $RT_K, RD_K$ | EM (%) ↑ | | | F1 (%) ↑ | | |
|---|---|---|---|---|---|---|
| | **DIG-R** | $\gamma$-**Order** | $p$-value ↓ | **DIG-R** | $\gamma$-**Order** | $p$-value ↓ |
| $4, 4$ | $31.23 \pm 1.13$ | $30.83 \pm 1.16$ | ~~0.102~~ | $32.90 \pm 0.96$ | $33.17 \pm 1.09$ | ~~0.241~~ |

# 7 DISCUSSION AND CONCLUSION

We introduced **Directed Information $\gamma$-covering** as a self-organizing principle for context engineering. By leveraging directed information to capture asymmetric predictive relations among chunks, we defined $\gamma$-covering as a query-agnostic structure that both preserves information (soundness) and enforces non-redundancy (diversity). Through its connection to submodular set cover, the greedy algorithm inherits $(1 + \ln n)$ and $(1 - 1/e)$ approximation guarantees, while admitting both dynamic and static variants for different computational trade-offs.

Our empirical study across reranking, compression, and system prompt selection demonstrates *consistent improvements over BM25*, a highly competitive retrieval baseline. Although our experiments are still limited in scope, the results suggest that $\gamma$-covering is particularly effective when forced to make *hard decisions*, such as discarding gold facts under hard compression or retaining only a single chunk for system prompts. These are precisely the settings where our framework provides a principled safeguard.

At the same time, we acknowledge that our experiments do not yet establish clear superiority over query-dependent PMI. We attribute this partly to the choice of dataset: HotpotQA contains distractors, which is query-dependent, but little true redundancy, whereas redundancy is abundant in real-world applications where $\gamma$-covering is expected to shine. Plus, PMI is query-dependent and incurs substantial online computational cost, whereas $\gamma$-covering operates offline and its cost can be amortized. As future work, we plan to extend evaluation to redundancy-rich settings such as multi-document QA and long-form summarization, where $\gamma$-covering is expected to shine. By reducing redundancy and stabilizing context under strict budgets, $\gamma$-covering has the potential to lower inference costs and make LLM pipelines more reliable and efficient.

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

# A APPENDIX

## A.1 PROOF OF PMI COUPLING BOUNDS

*Proof of Lemma 3.1.* By the chain rule for MI, $I(q; C_j) = I(q; C_i) + I(q; C_j \mid C_i) - I(q; C_i \mid C_j)$. Also, $0 \leq I(q; C_j | C_i) = H(C_j | C_i) - H(Cj | q, Ci) \leq H(C_j | C_i)$ and symmetrically $0 \leq I(q; C_i | C_j) \leq H(C_i | C_j)$, the inequality follows. □

## A.2 PROOF OF ESTIMATOR CONSISTENCY

*Proof of Theorem 3.2.* Fix a chunk pair $(C_i, C_j)$ and write $C_j = (y_1, \ldots, y_T)$ with $T = T_j$ for brevity. Let the (ideal) reference distribution be $p^\star$ and the parametric model be $p_\theta$. For each token position $t$, define the *per-token log-ratio scores*

$$s_t^\star := \log p^\star(y_t \mid y_{<t}, C_i^{\leq t}) - \log p^\star(y_t \mid y_{<t}), \qquad s_t^\theta := \log p_\theta(y_t \mid y_{<t}, C_i^{\leq t}) - \log p_\theta(y_t \mid y_{<t}).$$

By definition of $\hat{w}_{i \to j}$, we have

$$\hat{w}_{i \to j} = \frac{1}{T} \sum_{t=1}^{T} \left( \log p_\theta(y_t \mid y_{<t}, C_i^{\leq t}) - \log p_\theta(y_t \mid y_{<t}) \right) = \frac{1}{T} \sum_{t=1}^{T} s_t^\theta.$$

Moreover, by the standard identity for directed information(Massey, 1990),

$$I^\star(C_i^{\leq t}; y_t \mid y_{<t}) = \mathbb{E}_{p^\star}[s_t^\star \mid y_{<t}],$$

and hence the *directed-information rate* appearing in the theorem equals

$$\frac{1}{T}\sum_{t=1}^{T} I^{\star}(C_i^{\leq t}; y_t \mid y_{<t}) \;=\; \frac{1}{T}\sum_{t=1}^{T} \mathbb{E}_{p^{\star}}\!\left[s_t^{\star} \mid y_{<t}\right].$$

We now decompose the target deviation by a triangle inequality into an *approximation* term and a *stochastic* term:

$$\left|\hat{w}_{i \to j} - \frac{1}{T}\sum_{t=1}^{T} I^{\star}(C_i^{\leq t}; y_t \mid y_{<t})\right| = \left|\frac{1}{T}\sum_{t=1}^{T} s_t^{\theta} \;-\; \frac{1}{T}\sum_{t=1}^{T} \mathbb{E}_{p^{\star}}\!\left[s_t^{\star} \mid y_{<t}\right]\right|$$

$$\leq \underbrace{\left|\frac{1}{T}\sum_{t=1}^{T}\left(s_t^{\theta} - s_t^{\star}\right)\right|}_{\text{(A) approximation error}} + \underbrace{\left|\frac{1}{T}\sum_{t=1}^{T}\left(s_t^{\star} - \mathbb{E}_{p^{\star}}\!\left[s_t^{\star} \mid y_{<t}\right]\right)\right|}_{\text{(B) stochastic error}}.$$

**(A) Approximation error bound.** By Assumption (A1), we have a uniform log-likelihood approximation error

$$\sup_z \left|\log p_\theta(z) - \log p^{\star}(z)\right| \;\leq\; \varepsilon.$$

Applying this with $z = (y_t \mid y_{<t}, C_i^{\leq t})$ gives

$$\left|\log p_\theta(y_t \mid y_{<t}, C_i^{\leq t}) - \log p^{\star}(y_t \mid y_{<t}, C_i^{\leq t})\right| \leq \varepsilon,$$

Similarly, with $z = (y_t \mid y_{<t})$ gives

$$\left|\log p_\theta(y_t \mid y_{<t}) - \log p^{\star}(y_t \mid y_{<t})\right| \leq \varepsilon.$$

Hence, by the triangle inequality for differences of these two terms,

$$\left|s_t^{\theta} - s_t^{\star}\right| = \left|\left(\log p_\theta(\cdot \mid y_{<t}, C_i^{\leq t}) - \log p_\theta(\cdot \mid y_{<t})\right)\right.$$
$$\left. - \left(\log p^{\star}(\cdot \mid y_{<t}, C_i^{\leq t}) - \log p^{\star}(\cdot \mid y_{<t})\right)\right|$$
$$\leq 2\varepsilon.$$

Therefore,

$$\left|\frac{1}{T}\sum_{t=1}^{T}\left(s_t^{\theta} - s_t^{\star}\right)\right| \;\leq\; \frac{1}{T}\sum_{t=1}^{T} 2\varepsilon \;=\; 2\varepsilon.$$

**(B) Stochastic error bound.** Define the martingale difference sequence

$$\xi_t \;:=\; s_t^{\star} - \mathbb{E}_{p^{\star}}\!\left[s_t^{\star} \mid y_{<t}\right], \qquad t = 1, \ldots, T,$$

with respect to the filtration $\mathcal{F}_t = \sigma(y_{\leq t}, C_i^{\leq t})$. By construction, $\mathbb{E}_{p^{\star}}[\xi_t \mid \mathcal{F}_{t-1}] = 0$. Assumption (A2) states per-token losses are sub-Gaussian; since $s_t^{\star}$ is a *difference* of two such log-likelihood terms, $\xi_t$ is also sub-Gaussian (with some proxy variance parameter $\sigma^2$). Hence, by the Azuma–Hoeffding (see, e.g., Boucheron et al. (2013)) inequality for martingale differences,

$$\mathbb{P}\!\left(\left|\frac{1}{T}\sum_{t=1}^{T}\xi_t\right| \geq u\right) \;\leq\; 2\exp\!\left(-\frac{c\,T\,u^2}{\sigma^2}\right) \quad \text{for all } u > 0,$$

for a universal constant $c > 0$. Equivalently,

$$\frac{1}{T}\sum_{t=1}^{T}\left(s_t^{\star} - \mathbb{E}_{p^{\star}}\!\left[s_t^{\star} \mid y_{<t}\right]\right) \;=\; O_{\mathbb{P}}\!\left(\frac{1}{\sqrt{T}}\right).$$

**Conclusion.** Combining (A) and (B), we obtain

$$\left| \hat{w}_{i \to j} - \frac{1}{T} \sum_{t=1}^{T} I^\star(C_i^{\leq t}; y_t \mid y_{<t}) \right| \;\leq\; 2\varepsilon \;+\; O_{\mathbb{P}}\Big(\frac{1}{\sqrt{T}}\Big).$$

$\square$

In practice, autoregressive LMs operate with a finite context window. Assuming truncation error is negligible (A3), the same consistency guarantee holds with an added $\delta_{trunc}$ term.

### A.3 PROOF OF SAFE PRUNING

*Proof of Theorem 3.3.* Work on the high-probability event $\mathcal{E}$ where all four estimation bounds hold. By Lemma 3.1 and triangle bounds,

$$\begin{aligned}
\widehat{\mathrm{PMI}}(q; C_j) \leq \mathrm{PMI}^\star(q; C_j) + \delta_j \;&\leq\; \mathrm{PMI}^\star(q; C_i) + H^\star(C_j) - \mathrm{DI}_{i \to j} + \delta_j \\
&\leq\; \big(\widehat{\mathrm{PMI}}(q; C_i) + \delta_i\big) + \big(\hat{H}(C_j) + \eta_j\big) - \big(\hat{w}_{i \to j} - \epsilon_{ij}\big) + \delta_j \\
&=\; \widehat{\mathrm{PMI}}(q; C_i) + \hat{H}(C_j) - \hat{w}_{i \to j} + (\delta_i + \delta_j + \eta_j + \epsilon_{ij}).
\end{aligned}$$

Since $\mathbb{P}(\mathcal{E}) \geq 1 - \alpha$, the claim follows. $\square$

### A.4 PROOF OF SOUNDNESS OF $\gamma$-REPRESENTATIVES

*Proof of Theorem 3.5.* By lemma 3.1,

$$\mathrm{PMI}(q; C_j) \leq \mathrm{PMI}(q; Ci) + H(C_j \mid C_i)$$

We also have $H(C_j \mid C_i) \leq H(C_j) + \mathrm{DI}_{i \to j}$, hence if $i$ $\gamma$-covers $j$ we have $\mathrm{PMI}(q; C_j) \leq \mathrm{PMI}(q; C_i) + \gamma$ in the ideal ($p^\star$) case. Incorporating estimation, Theorem 3.3 gives the high-probability bound

$$\mathrm{PMI}(\hat{q}; C_j) \leq \mathrm{PMI}(\hat{q}; C_i) + \hat{H}(C_j) - \hat{w}_{i \to j} + (\delta_i + \delta_j + \eta_j + \epsilon_{ij}),$$

and under the $\gamma$-cover test $\hat{w}_{i \to j} \geq \hat{H}(C_j) - \gamma$ this becomes

$$\mathrm{PMI}(\hat{q}; C_j) \leq \mathrm{PMI}(\hat{q}; C_i) + \gamma + (\delta_i + \delta_j + \eta_j + \epsilon_{ij}) \tag{1}$$

Now order the items of $U \setminus S$ arbitrarily as $j_1, \ldots, j_m$ and apply the chain rule / subadditivity for mutual information to expand $I(q; U)$ by adding $C_{j_t}$ one at a time. Each increment is $I(q; C_{j_t} \mid S \cup j_{<t}) \leq \mathrm{PMI}(q; C_{j_t})$; substitute the bound (1) with its representative $i \in S$, and sum over $t = 1, \ldots, m$. This yields the claimed bound relative to $I(q; S)$ with the additive slacks. The uniform-slack corollary follows immediately. $\square$

### A.5 PROOF OF DIVERSITY MARGIN

*Proof of Proposition 3.2.* By definition of a representative set for a $\gamma$-cover, for any distinct $i \neq j \in S$ neither $i$ $\gamma$-covers $j$ nor $j$ $\gamma$-covers $i$. Thus $\mathrm{DI}_{i \to j} < H(C_j) - \gamma$ and $\mathrm{DI}_{j \to i} < H(C_i) - \gamma$. Using $H(C_j \mid C_i) = H(C_j) - I(C_i; C_j) \leq H(C_j) - \mathrm{DI}_{i \to j}$ (and symmetrically), both conditional entropies exceed $\gamma$, establishing the claim. $\square$

### A.6 PROOF OF DIFFUSION ALGORITHM COMPLEXITY AND TERMINATION

In Algorithm 2, computing all pairwise $\hat{w}_{i \to j}$ requires $O(M^2 T)$ forward passes, where $M$ is the candidate pool size and $T$ the average chunk length. The diffusion step costs $O(M^2)$, reducible to $O(MK)$ if each node retains only its top-$K$ predictive neighbors. Iterating the update

$$r^{(t+1)} = \alpha r^{(0)} + (1 - \alpha) P^\top r^{(t)}$$

yields an affine contraction with constant $1 - \alpha$, guaranteeing a unique fixed point and geometric convergence by the Banach fixed-point theorem (Page et al., 1999). Theorem A.1 summarizes the computational and convergence properties.

**Theorem A.1** (Algorithmic complexity and termination). Consider DIG-R with neighborhood size $M$ and chunk length $\leq T$.

1. Edge computation costs $O(M^2 T)$ forward tokens, batchable across GPUs.

2. Propagation costs $O(M^2)$ (or $O(MK)$ for sparse $K$-Nearest Neighbor edges).

3. With damping $\alpha \in (0, 1)$, the propagation step is a contraction mapping and converges to a unique fixed point in $O(\log(1/\varepsilon))$ iterations.

**Proposition A.1** (Convergence of the damped propagation). Let $P \in \mathbb{R}^{M \times M}$ be column-stochastic (each column sums to 1 and entries are nonnegative), fix $\alpha \in (0, 1)$, and define

$$F(r) := \alpha\, r^{(0)} + (1 - \alpha)\, P^\top r, \qquad r \in \mathbb{R}^M.$$

Then $F$ is a contraction mapping on $(\mathbb{R}^M, \|\cdot\|_1)$ with contraction factor $(1 - \alpha)$, hence admits a unique fixed point $r^\star$ and the iteration $r^{(t+1)} = F(r^{(t)})$ converges to $r^\star$ at a geometric rate:

$$\|r^{(t)} - r^\star\|_1 \leq (1 - \alpha)^t \|r^{(0)} - r^\star\|_1,$$

so that $\|r^{(t)} - r^\star\|_1 \leq \varepsilon$ after $t = O(\log(1/\varepsilon))$ steps.

*Proof sketch.* For any $r, u \in \mathbb{R}^M$,

$$\|F(r) - F(u)\|_1 = (1 - \alpha)\, \|P^\top (r - u)\|_1.$$

Since $P$ is column-stochastic, $P^\top$ is row-stochastic and is a nonexpansive linear operator in $\ell_1$:

$$\|P^\top v\|_1 \leq \|v\|_1 \quad \text{for all } v \in \mathbb{R}^M.$$

Therefore,

$$\|F(r) - F(u)\|_1 \leq (1 - \alpha)\, \|r - u\|_1,$$

so $F$ is a contraction with constant $(1 - \alpha) < 1$. By the Banach fixed-point theorem, $F$ has a unique fixed point $r^\star$ and the Picard iteration $r^{(t+1)} = F(r^{(t)})$ converges to $r^\star$ with

$$\|r^{(t)} - r^\star\|_1 \leq (1 - \alpha)^t \|r^{(0)} - r^\star\|_1.$$

Solving $(1 - \alpha)^t \|r^{(0)} - r^\star\|_1 \leq \varepsilon$ yields $t \geq \frac{\log\left(\|r^{(0)} - r^\star\|_1 / \varepsilon\right)}{\log(1/(1-\alpha))} = O(\log(1/\varepsilon))$. $\qquad \square$

