# OpenReview forum: "Directed Information $\gamma$-covering: An Information-Theoretic Framework for Context Engineering"
_ICLR.cc/2026/Conference — Submitted to ICLR 2026_

### Official Review · Reviewer_eAMx · 2025-10-26

**Soundness:** 4
**Presentation:** 3
**Contribution:** 2
**Rating:** 4
**Confidence:** 3

**Summary:**

This paper introduces a novel and principled information-theoretic framework for context engineering, called Directed Information $\gamma$-Covering. The core idea is to leverage the asymmetric, predictive nature of Directed Information (DI) to build a query-agnostic "cover set" of context chunks. The authors formally define a $\gamma$-cover as a subset of chunks $S$ such that any chunk $j$ not in $S$ is "covered" by some chunk $i$ in $S$, meaning $DI_{i\rightarrow j} \ge H(C_j) - \gamma$. This criterion guarantees that $C_i$ can represent $C_j$ up to a $\gamma$-bit information loss.

The authors provide a crucial theoretical bridge (Lemma 3.1) linking this query-agnostic DI structure to query-dependent Pointwise Mutual Information (PMI), which is a common measure of relevance in RAG. They formulate the selection of this cover set as a submodular set cover problem, for which they propose a greedy algorithm (Algorithm 1) that inherits a $(1-1/e)$ approximation guarantee. The framework is supported by theoretical guarantees for information preservation (soundness, Theorem 3.5) and non-redundancy (diversity, Proposition 3.2). Empirically, the authors apply this framework to context compression, system prompt selection, and reranking on HotpotQA. The results show that $\gamma$-covering provides clear advantages in "hard-decision" regimes (e.g., high compression, single-prompt selection) and offers modest, consistent gains in reranking over a strong BM25 baseline, all while being a query-agnostic method that can be computed offline.

**Strengths:**

- The application of Directed Information to context engineering is, to my knowledge, novel. Using the asymmetry of DI ($DI_{i\rightarrow j}$ vs. $DI_{j\rightarrow i}$) to model predictive relationships is a clever and fine-grained approach, moving beyond more common symmetric measures like MI or perplexity. The paper is theoretically dense and well-constructed.

-  The authors do an excellent job of building the framework from the ground up. The bridge from query-dependent PMI to the query-agnostic DI covering criterion (Lemma 3.1 and its corollaries) is the theoretical cornerstone that justifies the entire offline approach. This is a significant contribution.

- Connecting the problem to submodular set cover is a major strength. This immediately allows the authors to import strong, well-understood approximation guarantees (Theorem 3.4) for their greedy selection algorithm. The additional formal guarantees for soundness (information preservation up to an additive slack) and diversity (an information-theoretic margin between representatives) provide a solid, principled foundation for why this method should work.

**Weaknesses:**

-  The theoretical guarantees (e.g., Theorem 3.5) are derived with respect to the true distributions ($p^{\*}$) and information measures ($DI^\*, H^\*$). The practical implementation, however, relies on empirical estimators from a model $p_\theta$.
   - The "Safe Pruning" guarantee (Theorem 3.3) shows that estimation errors from all components ($\delta_i, \delta_j, \eta_j, \epsilon_{ij}$) accumulate.
   - Vacuous Soundness Bound? This culminates in the final soundness bound (Theorem 3.5), where the total information slack is $|U \setminus S|(\gamma + 3\bar{\delta})$. In a setting with many chunks to be compressed (large $|U \setminus S|$), this additive error bound could become very loose or even vacuous, especially if the uniform estimation error $\bar{\delta}$ is non-trivial. The paper lacks a discussion or empirical characterization of this crucial estimation error, which is the primary link between the theory and the practice.

- The entire framework is parameterized by the tolerance $\gamma$. This parameter seems to be the most important hyperparameter, as it directly controls the trade-off between compression (cover size $|S|$) and information loss. How is $\gamma$ chosen? Is it tuned per task, per dataset, or set globally? How sensitive are the results to this choice? A small $\gamma$ could result in $|S| \approx |U|$ (no compression), while a large $\gamma$ would lead to high compression but potentially invalid soundess guarantees. This practical aspect is critical to the method's usability but is not discussed.

- The authors are honest about the results, but they are not a strong endorsement. The method only clearly outperforms the PMI baseline in "hard compression" regimes. In "soft compression" (Tables 1 & 2), the query-dependent PMI baseline is significantly better. The authors' defense (HotpotQA has query-dependent distractors but low redundancy) is plausible, but it also suggests that the practical benefits of this query-agnostic method may be limited to specific dataset types (high redundancy) or specific, "hard" tasks. This trade-off between offline computation and online performance needs to be explored more thoroughly.

- Disconnect of DIG-R (Reranking): Algorithm 2 (DIG-R) feels somewhat disconnected from the core $\gamma$-covering framework. It uses the raw DI weights ($\hat{w}_{i\rightarrow j}$) in a PageRank-style diffusion, not the $\gamma$-cover sets or the greedy algorithm (Algorithm 1). It's a graph-based reranker, but it doesn't seem to leverage the key theoretical contributions (set cover, soundness guarantees). The ablation in Table 6, which shows that a simple "$\gamma$-order" (presumably from Algorithm 1?) is not statistically different from DIG-R, further complicates this. The paper would be stronger if it either justified this connection better or focused the empirical section on applications that directly use the $\gamma$-cover set $S$.

**Questions:**

- On the Magnitude of Estimation Errors: Following Theorem 3.5, the soundness bound $I(q;U) \le I(q;S) + |U \setminus S|(\gamma + 3\bar{\delta})$ seems highly dependent on the number of compressed items. Could you provide an empirical analysis of the estimation slack $\bar{\delta}$ on a dataset like HotpotQA? How large is this term in practice, and at what corpus size $|U|$ would this error term dominate $\gamma$, potentially weakening the guarantee?

- On Selecting $\gamma$: How was the tolerance $\gamma$ selected for the experiments in Tables 1 and 2? Was it tuned on a validation set? Could you provide a sensitivity analysis showing how the cover size $|S|$ and the downstream EM/F1 scores vary with different choices of $\gamma$?

- On Scalability: The offline $O(M^2)$ computation to build the DI graph is a bottleneck for very large $M$. Have you considered or explored methods to approximate the $\gamma$-cover without computing all pairwise interactions? For example, could techniques from LSH or other similarity search methods be used to find a candidate set of $(i, j)$ pairs for which $DI_{i\rightarrow j}$ is likely to be high?

- On the MI vs. DI Ablation (Table 4): The finding that DI outperforms MI is very interesting. The MI criterion $I(i;j) \ge H(C_j) - \gamma$ is a stronger condition than the DI one (since $I \ge DI$). Intuitively, one might guess that a stronger condition would select "better" or more robust representatives. Why does it perform worse? My hypothesis is that it inefficiently "over-covers" (e.g., it might prefer a symmetric pair $(i, j)$ where $DI_{i\rightarrow j}$ and $DI_{j\rightarrow i}$ are both high, when a single asymmetric chunk $k$ with high $DI_{k\rightarrow i}$ and $DI_{k\rightarrow j}$ would have been a more efficient cover). Does this match your intuition?

---

> ### Author Response · Authors · 2025-11-24
>
> > HotpotQA has query-dependent distractors but low redundancy.
>
> Thanks for pointing this out. We agree that HotpotQA is not the most suitable dataset for this purpose. To address this, we substantially revised our experiments by **augmenting** HotPotQA with **paraphrases** and **summaries**. These simulate two common real-world sources of redundancy:
> 1. multiple versions or editions of the same underlying document (paraphrases), and
> 1. brief mentions of a topic in related contexts (summaries).
>
> With this augmented dataset, we now observe significant gains over PMI:
> *  up to **10-point improvements in EM and F1**, and
> *  up to **20-point improvements in recall** of the golden supporting facts.
>
> Please see the first three subsections of the **Response to All Reviewers: Additional Experiments** for details.
>
> > Could you provide an empirical analysis of the estimation slack $\bar{delta}$  on a dataset like HotpotQA?
>
> Thanks for the question. We conducted empirical study and show that the average stalk is consistently below 2% of PMI(q;c), indicating that 𝛿 is small and leaving ample room for the |U/S| factor. See Empirical Study on $\bar{delta}$ in the “Response to All Reviewers: Additional Experiments” for details.
>
> > How was the tolerance $\gamma$ selected for the experiments in Tables 1 and 2?
>
> We first note that it is **not necessary to choose a single global value of $\gamma$**. In practice, $\gamma$ can be set per question, which provides additional flexibility. Importantly, selecting a query-dependent $\gamma$ does _not_ shift any computation to runtime: all DI values are still computed offline, and at inference time we simply retrieve the precomputed scores and use the chosen $\gamma$ to determine when to stop.
>
> Furthermore, the algorithm can be terminated either
> 1. by a threshold on $\gamma$, or
> 2. by a threshold on the size of the selected context set,
>
> 2 is often more practical, as users may wish to explicitly trade off performance and cost. In fact, both experiments in Table 1 and 2 use a size-based stopping criterion.
>
> > Could you provide a sensitivity analysis showing how the cover size $|S|$ and the downstream EM/F1 scores vary with different choices of $\gamma$?
>
> We conducted additional experiments and observed **an approximately linear relationship** between $\gamma$ and the size of the selected context set. This further supports using the number of selected contexts as a threshold, since increasing $\gamma$ does not produce unpredictable or uneven scaling in quality. The experiments also illustrate how to select $\gamma$ to achieve specific levels of recall, EM, and F1. Please see the _Empirical Study on $\gamma$_ section in the **Response to All Reviewers: Additional Experiments** for details.
>
> > Have you considered or explored methods to approximate the $\gamma$-cover without computing all pairwise interactions?
>
> Thank you for the insightful question. We have examined this scenario, but since directed information is **not a metric** (it does not satisfy the triangle inequality), we cannot use metric-based pruning tricks such as computing only ${\rm DI}_{s\rightarrow c}$ for $s\in S$, the current representative set. An LHS-based solution may suffer from the same issue. That said, a few promising directions remain:
> 1. Prefix sharing: our NLL-based estimator is additive, so if two chunks $c_i$ and $c_j$ share prefixes, computation can be reused.
> 1. Use small models: as shown in our ablation study (see _Ablation on Model Sizes and Model Families_ in the **Response to All Reviewers: Additional Experiments**), our approach is **not sensitive to the choice of LLM**. Therefore, using a small model is a practical way to reduce computation cost without sacrificing performance.
> 1. Parallelization: although the total FLOPs remain, the computation is fully parallelizable, which can reduce latency, if not total cost.
>
> We plan to investigate these optimizations in future work.
>
> > Why does [MI] perform worse?
>
> Including summaries in the dataset helped clarify why DI outperforms MI. Because MI is symmetric, $I(c;s) = I(s;c)$, where $c$ is a context chunk and $s$ is its summary. In contrast, DI is directional: ${\rm DI}_{s\rightarrow c}$ is smaller because the context contains more detailed information than its summary. In other words, MI cannot distinguish between $c$ and $s$, whereas DI can.
>
> We measured the fraction of summaries selected into the representative set and found that MI selects summaries at a much higher rate than DI. Because some summaries lack the necessary level of detail for answering the question, this leads to a substantial drop in accuracy when using MI.
>
> k|EM DI|EM MI|F1 DI|F1 MI|Summary Ratio DI|Summary Ratio MI
> -|-|-|-|-|-|-
> 1|33.7|30.7|38.42|36.23|2.56|12.23
> 2|41.7|39.51|46.59|44.84|7.12|23.56
> $s$|44.25|42|48.84|47.18|8.96|26.98
> $s+1$|50.35|47.31|54.42|52.18|15.42|38.05
> $s+2$|53.62|51.19|57.88|55.7|23.97|48.19
> $s+3$|55.06|53.24|59.36|57.97|33.74|57.08

---

> ### Comment · Reviewer_eAMx · 2025-11-26
>
> Thank you for the response. I'll maintain my current score.

---

> > ### Author Response · Authors · 2025-11-29
> >
> > Thanks for taking the time to read our rebuttal. If you’re open to it, we would greatly appreciate hearing a bit more about which concerns remain

---

### Official Review · Reviewer_6c65 · 2025-10-31

**Soundness:** 2
**Presentation:** 3
**Contribution:** 3
**Rating:** 4
**Confidence:** 3

**Summary:**

The authors formulate context/prompt selection as a γ-cover problem and propose Directed Information (DI) γ-covering, an information-theoretic framework for query-agnostic redundancy-aware “context diet” framework for LLMs. Unlike existing methods based on Pointwise Mutual Information (PMI), DI captures causal and asymmetric dependencies between text chunks, enabling self-organizing selection and compression of contextual information for LLMs. In this framework, DI is approximated by the negative log-likelihood of the LLM, and one chunk is said to γ-covers another if it can predict the other chunk within a margin of γ bits of uncertainty. The authors provide theoretical guarantees for γ-covering and empirically argue its effectiveness on the HotpotQA dataset, showing consistent improvements over PMI in context compression and single-slot prompt selection, and over vanilla BM25 in reranking tasks.

**Strengths:**

S1. The authors use an information-theoretic perspective to tackle the important problem of context and prompt engineering for LLMs.

S2. The authors formulate context/prompt selection as a γ-cover problem and propose Directed Information (DI) γ-Covering.

S3. The authors provide theoretical guarantees to support the effectiveness of the proposed DI γ-Covering.

**Weaknesses:**

W1. The authors should clarify the suitability of HotpotQA as the evaluation benchmark. Specifically, it is unclear whether HotpotQA is an appropriate dataset for evaluating the effectiveness of the query-agnostic DI γ-Covering framework. HotpotQA consists of two gold supporting paragraphs and eight distractors per query, making it inherently query-dependent. Since DI γ-Covering operates without query information, it is not evident how increasing DI among paragraphs can ensure that the two relevant supporting paragraphs are selected. The paper would benefit from either a clearer justification for using HotpotQA or additional experiments on datasets where query-agnostic selection is more natural.

W2. The experimental setup lacks clarity.

W2.1 In Section 4, the definition of the compression size is confusing. The authors define s as the number of gold supporting facts and d as the number of distractors. However, given that s=2 and d=8 in HotpotQA, some of the reported compression sizes s-2 become zero. This makes the reported settings (i.e., s+d-1, s+d-2, s, s-1, s-2) unintuitive and difficult to interpret.

W2.2 The authors should clarify which LLM was used in the context compression and system prompt selection experiments.

W3. The experiments should be more comprehensive.

W3.1 The authors should include a direct evaluation of the selection quality by reporting the recall of gold supporting facts (i.e., the percentage of annotated gold paragraphs retained after applying DI γ-covering). This would provide a clearer measure of how well the method preserves relevant evidence, complementing the end-to-end EM/F1 scores that reflect final answer quality.

W3.2 In Table 2, the authors should report EM and F1 scores when the LLM is given only the gold supporting paragraphs. This upper-bound accuracy comparison would contextualize the performance of γ-covering and help quantify its effectiveness relative to the ideal scenario.

**Questions:**

Please refer to W1, W2, and W3.

---

> ### Author Response · Authors · 2025-11-24
>
> > W1. … clarify the suitability of HotpotQA as the evaluation benchmark.
>
> Thanks for the question. Although HotPotQA does not naturally contain the level of redundancy found in real-world settings, it is a **multihop QA dataset**—making it well suited for demonstrating our **diversity** guarantees (one of the key strengths of our approach). In addition, it provides **explicitly labeled supporting facts**, which are essential for **recall analysis**.
>
> That said, we agree that HotpotQA is not the most suitable dataset for this purpose. To address this, we substantially revised our experiments by **augmenting** HotPotQA with **paraphrases** and **summaries**. These simulate two common real-world sources of redundancy:
> 1. multiple versions or editions of the same underlying document (paraphrases), and
> 2. brief mentions of a topic in related contexts (summaries).
>
> With this augmented dataset, we now observe significant gains over PMI:
> *  up to **10-point improvements in EM and F1**, and
> *  up to **20-point improvements in recall** of the golden supporting facts.
>
> Please see the first three subsections of the **Response to All Reviewers: Additional Experiments** for details.
>
> > W2.1 This makes the reported settings (i.e., $s+d-1$, $s+d-2$, $s$, $s-1$, $s-2$) unintuitive and difficult to interpret.
>
> Also thanks for pointing out the overlap between hard compression and system-prompt selection in our initial experimental design. Let $s$ denote the number of supporting facts in the ground truth. We now **redefine** context compression as reducing the number of context chunks to any value $\ge s$, whereas system-prompt selection refers to reducing the number of chunks to $<s$. This removes the overlap between the two use cases. With the revamped experiments, our approach now shows superior performance in both settings.
>
> > W2.2 The authors should clarify which LLM was used
>
> We further clarify that we use `Llama-3.2-3B-Instruct` to compute DI and PMI. In the revamped experiments, we also include ablations across different model sizes and model families, which show that our approach is not sensitive to the choice of LLM. Please see _Ablation on Model Sizes and Model Families_ in the **Response to All Reviewers: Additional Experiments** for details.
>
> > W3.1 … reporting the recall of gold supporting facts.
>
> Thank you for the valuable suggestion. We incorporated recall into our revamped experiments and observed up to a **20-point improvement**. Note that recall is computed only on the original supporting facts and does **not count paraphrases or summaries**. Please see _Results and Findings (with Recall)_ in the **Response to All Reviewers: Additional Experiments** for details.
>
> > W3.2 The authors should report EM and F1 when the LLM is given only the gold supporting paragraphs.
>
> We conducted this experiment and report the numbers below.
>
> EM|F1
> -|-
> 58.02$\pm$0.89|63.26$\pm$0.58
>
>
> We note that with $\gamma=0.5$, our approach achieves 57.61 EM and 61.38 F1, which is close to the best possible results listed above. See _Empirical Study on $\gamma$_ in the **Response to All Reviewers: Additional Experiments** for details.

---

> > ### Comment · Reviewer_6c65 · 2025-11-28
> >
> > Thank you for the detailed response. However, I still have fundamental concerns regarding W1 of my initial review. I will maintain my score.
> >
> > I do not yet understand why conducting context compression via the γ-covering algorithm improves QA accuracy in the HotpotQA setting (even with the augmented dataset). It remains unclear whether extending HotpotQA in that manner is the right approach for answering my concerns. γ-covering is a query-agnostic framework that finds the informative representative chunks: it builds a directed information graph over chunks and then uses a greedy set-cover procedure to select a subset of chunks whose information γ-covers the remaining chunks. In contrast, in HotpotQA the goal is to select chunks based on their relevance to the query. From the perspective of a query-agnostic DI graph, both supporting paragraphs and distractor paragraphs are simply coherent text chunks, and the method cannot directly distinguish which ones are “Gold Facts” versus “Distractors.” Without access to the query signal, how does maximizing γ-coverage over context chunks lead the algorithm to preferentially retain query-relevant supporting facts rather than distractor paragraphs?

---

> > > ### Author Response · Authors · 2025-11-29
> > >
> > > The additional experiment is intended to highlight a typical failure mode of query-dependent methods. When multiple redundant supporting facts exist, a query-dependent approach will tend to select all of them, consuming the limited context budget and leaving insufficient room for other necessary supporting evidence. This is precisely why PMI performs worse on the augmented dataset with paraphrases and summaries.
> > >
> > > Taken together, the added experiments—along with those already in the paper—illustrate both **where our approach excels** (in redundancy-rich settings) and its continued **effectiveness even in an adversarial scenario**, such as system-prompt selection on HotPotQA, where query-dependence favors PMI. We believe this constitutes a sufficient and informative empirical study for a contribution whose primary novelty is theoretical.
> > >
> > > We hope the reviewers will evaluate the work as a theoretical contribution supported by sufficient empirical evidence to demonstrate its effectiveness and guide future research. We are sincerely grateful for your thoughtful engagement, and we would be happy to clarify anything further.

---

### Official Review · Reviewer_qvkH · 2025-10-31

**Soundness:** 3
**Presentation:** 3
**Contribution:** 3
**Rating:** 4
**Confidence:** 3

**Summary:**

This paper addresses the problem of context selection and compression in large-scale language models. It proposes a Directed Information (DI) γ-covering approach, which efficiently selects, compresses, and diversifies context without losing essential information. The method introduces Directed Information to capture asymmetric predictive relationships between context chunks, forming a query-agnostic, self-organizing framework. Additionally, the authors design a greedy algorithm and provide theoretical guarantees on information retention, diversity, and approximation. Experimental results show that DI γ-covering outperforms traditional baseline methods in tasks like context compression and system prompt selection. This paper provides an effective solution for context engineering problems in large language models. The paper is logically sound, but there are some limitations in the experimental section.

**Strengths:**

1.	This paper offers a novel perspective by introducing Directed Information (DI) into context engineering tasks. The γ-covering method proposed provides a query-agnostic, self-organizing framework that offers a fresh approach to the retrieval-augmented generation (RAG) paradigm, which has traditionally relied on query-dependent reordering.

2.	The paper is theoretically robust, modeling asymmetric predictive relationships between context chunks via Directed Information (DI). It presents solid theoretical foundations and proofs, including guarantees on approximation and method soundness, making the entire framework reliable.

3.	The paper introduces a query-agnostic context selection method with DI γ-covering, reducing the need for expensive online computations. This significantly enhances computational efficiency, making the method scalable for large datasets and practical applications.

**Weaknesses:**

1.	The experiments are mainly focused on the HotpotQA dataset, which may not fully showcase the method's performance across other complex or diverse datasets. There is a lack of experimentation on multiple datasets (e.g., multi-document QA, summarization).Insufficient baselines. Comparisons focus mainly on PMI for ranking and BM25 for reranking, while several recent competitive methods are missing, which diminishes the strength of the empirical claims.

2.	The comparison of baseline methods is limited to PMI and BM25, with no comparison to more recent methods.

3.	It does not show visualizations comparing which context chunks are selected, merged, or removed by DI γ-covering versus PMI and other baselines.

**Questions:**

1.	Is the performance of this method sensitive to the choice of γ? How should γ be set or adjusted?

2.	Can the paper provide a visualization demonstrating the context chunks selected or removed by γ-covering compared to PMI and other baselines?

3.	The paper mentions that the method performs worse than PMI in soft compression. Could the authors provide specific examples where DI γ-covering made suboptimal choices in such scenarios and analyze why it happened?

---

> ### Author Response · Authors · 2025-11-24
>
> > HotpotQA dataset … may not fully showcase the method’s performance …
>
> Thanks for the feedback. We substantially revised our experiments by **augmenting** HotPotQA with **paraphrases** and **summaries**. These simulate two common real-world sources of redundancy:
> 1. multiple versions or editions of the same underlying document (paraphrases), and
> 1. brief mentions of a topic in related contexts (summaries).
>
> With this augmented dataset, we now observe significant gains over PMI:
> *  up to **10-point improvements in EM and F1**, and
> *  up to **20-point improvements in recall** of the golden supporting facts.
>
> Please see the first three subsections of the **Response to All Reviewers: Additional Experiments** for details.
>
> > The comparison of baseline methods is limited to PMI and BM25, with no comparison to more recent methods.
>
> We focus on PMI and BM25 because PMI is a recently proposed SOTA baseline (NAACL 2025) that is also grounded in information theory, making it the most directly comparable to our method. Moreover, PMI is a strong and challenging baseline because it is fully **query-dependent**, whereas our approach is query-agnostic. BM25 is included as a widely used and highly competitive retrieval baseline, providing a strong traditional comparison point.
>
> We are happy to consider additional baselines and would appreciate any specific recommendations the reviewer may have.
>
> > Is the performance of this method sensitive to the choice of $\gamma$? How should γ be set or adjusted?
>
> We first note that it is **not necessary to choose a single global value of $\gamma$**. In practice, $\gamma$ can be set per question, which provides additional flexibility. Importantly, selecting a query-dependent $\gamma$ does _not_ shift any computation to runtime: all $\rm DI$ values are still computed offline, and at inference time we simply retrieve the precomputed scores and use the chosen $\gamma$ to determine when to stop.
>
> Furthermore, the algorithm can be terminated either
> 1. by a threshold on $\gamma$, or
> 2. by a threshold on the size of the selected context set,
>
> 2 is often more practical, as users may wish to explicitly trade off performance and cost. In fact, most experiments in the paper use a size-based stopping criterion.
>
> We conducted additional experiments and observed **an approximately linear relationship** between $\gamma$ and the size of the selected context set. This further supports using the number of selected contexts as a threshold, since increasing $\gamma$ does not produce unpredictable or uneven scaling in quality. The experiments also illustrate how to select $\gamma$ to achieve specific levels of recall, EM, and F1. Please see the _Empirical Study on $\gamma$_ subsection in the **Response to All Reviewers: Additional Experiments** for details.
>
> > visualization demonstrating the context chunks selected or removed
>
> We created visuals at https://bit.ly/gamma-cover-visuals (shared anonymously). From these visuals, we observe that when the dataset contains _no_ redundancy, PMI tends to select supporting facts that are directly aligned with the question, whereas our approach prioritizes **diversity** and may therefore select facts that are less closely related to the specific query. However, when the dataset does contain redundancy, PMI tends to select the paraphrases and summaries of the supporting facts, while our approach does not, highlighting the robustness of our approach in redundancy-rich settings.

---

> > ### Comment · Reviewer_qvkH · 2025-11-27
> >
> > Thank you for your detailed response. Your explanations regarding the baseline comparisons and the additional clarifications on the visualization results have essentially addressed my questions. However, considering the overall contributions of the paper, I still maintain my original score.

---

> > > ### Author Response · Authors · 2025-11-29
> > >
> > > Thank you for your thoughtful follow-up and for noting that our additional explanations have “essentially addressed your questions.” We also sincerely appreciate your initial recognition that the contribution is “novel” and “theoretically robust.”
> > >
> > > That said, we would like to gently reiterate that the core contribution of our work is a **new theoretical framework with formal guarantees**. For theoretical work, it is standard practice to provide empirical evidence illustrating the framework’s effectiveness. We believe that both the experiments included in the paper and the additional experiments presented in the rebuttal offer a comprehensive set of results that validate the key claims and help guide future research directions.
> > >
> > > We hope the reviewers will evaluate the work as a theoretical contribution supported by sufficient empirical evidence to demonstrate its effectiveness and guide future research. We are sincerely grateful for your thoughtful engagement, and we would be happy to clarify anything further.

---

### Official Review · Reviewer_yKRm · 2025-11-01

**Soundness:** 3
**Presentation:** 3
**Contribution:** 3
**Rating:** 6
**Confidence:** 3

**Summary:**

This paper introduces Directed Information covering, a query-agnostic framework for context engineering. It leverages DI, an asymmetric measure of predictiveness between context chunks, to identify and remove redundancy. By formulating context selection as a gamma-cover problem, the authors propose a greedy algorithm that computes a "self-organizing" context structure offline, incurring no online cost. This method comes with provable guarantees for information preservation (soundness) and non-redundancy (diversity). Experiments on HotpotQA demonstrate that it significantly outperforms baselines in "hard-decision regimes," such as extreme context compression or single-prompt selection.

**Strengths:**

### Reduced Query Cost
The framework's primary strength is its novel approach to reducing online query costs by shifting computation offline. It pre-processes context by calculating the degree of predictive information between chunks, which minimizes search-time latency.

### Theoretical Soundness
The method is built on a strong theoretical foundation, offering provable guarantees for its performance, including information preservation and diversity

**Weaknesses:**

### Overhead in Dynamic Contexts
A major concern is the high computational overhead when new chunks are added or deleted. Adding even one new chunk would necessitate a costly $O(M)$ recalculation to update its DI relationships with all other chunks. Since retriever contexts in RAG systems often require frequent updates for information freshness, the framework's reliance on offline computation would necessitate costly recalculations of the DI relationships.


### Limited and Narrow Evaluation
The experimental validation is insufficient and relies almost exclusively on the HotpotQA dataset. The authors explicitly state that their sole evaluation dataset, HotpotQA, contains "relatively little redundancy". I think the framework was not tested in the "redundancy-rich settings" . To demonstrate that this method is generalizable and efficient, it must be validated across a more diverse range of QA and retrieval datasets.

**Questions:**

### Unaddressed Practicality of DI Estimation
The entire $O(M^2 T)$ offline computation depends on using a specific language model as a measurement tool. I think it's better to analyze how the choice of this model (e.g., its size or quality) impacts the resulting DI graph and final performance.

---

> ### Author Response · Authors · 2025-11-24
>
> > Overhead in dynamic context.
>
> Thank you for the insightful feedback. Since directed information is _not_ a metric (it does not satisfy the triangle inequality), we cannot use metric-based pruning tricks such as computing only ${\rm DI}_{s\rightarrow c}$ for $𝑠 \in 𝑆$, the current representative set. That said, a few promising directions remain:
> 1. Prefix sharing: our NLL-based estimator is additive, so if two chunks $c_i$ and $c_j$ share prefixes, computation can be reused.
> 1. Use small models: as shown in our ablation study (see _Ablation on Model Sizes and Model Families_ in the **Response to All Reviewers: Additional Experiments**), our approach is not sensitive to the choice of LLM. Therefore, using a small model is a practical way to reduce computation cost without sacrificing performance.
> 1. Parallelization: although the total FLOPs remain, the computation is fully parallelizable, which can reduce latency, if not total cost.
>
> We plan to investigate these optimizations in future work.
>
> > The framework was not tested in the “redundancy-rich” settings.
>
> Thank you for highlighting this. We substantially revised our experiments by **augmenting** HotPotQA with **paraphrases** and **summaries**. These simulate two common real-world sources of redundancy:
> 1. multiple versions or editions of the same underlying document (paraphrases), and
> 2. brief mentions of a topic in related contexts (summaries).
>
> With this augmented dataset, we now observe significant gains over PMI:
> *  up to **10-point improvements in EM and F1**, and
> *  up to **20-point improvements in recall** of the golden supporting facts.
>
> Please see the first three subsections of the **Response to All Reviewers: Additional Experiments** for details.
>
> > The … offline computation depends on using a specific language model as a measuring tool.
>
> We ran ablations across multiple model sizes and families, including `Llama-3.2-3B-Instruct`, `Llama-3.2-1B-Instruct`, and `Qwen-2.5-3B-Instruct`. The resulting EM and F1 scores were highly consistent across all models, indicating that our approach is **not sensitive to the choice of LLM** used for computing DI/PMI.
>
> See _Ablation on Model Sizes and Model Families_ in the **Response to All Reviewers: Additional Experiments**.

---

> > ### Comment · Reviewer_yKRm · 2025-11-28
> > **Thank you for the response**
> >
> > Thank you for the response and I have some concerns about that.
> >
> > 1. I think it does not fundamentally address the overhead in dynamic context problems. The solutions provided by the authors can lead to more optimized implementation of the method, but it inherently induce serious bottlenecks that prevent the method from utilization widely.
> >
> > 2. why the authors choose to augment HotpotQA on their own, rather than evluating own existing well-known datsets similar to HotpotQA. By using various kinds of existing datsets, the authors can strengthen their arguments more reliably.

---

> > > ### Author Response · Authors · 2025-11-29
> > >
> > > Thank you for clearly articulating your remaining concerns.
> > >
> > > > “it does not fundamentally address the overhead in dynamic context problems”
> > >
> > > As mentioned in our initial response, we view this as an important direction for future work. Our main contribution is a **novel theoretical framework** whose soundness is supported by formal guarantees. Designing a highly optimized implementation is primarily an engineering challenge that is somewhat orthogonal to the core theoretical advance. Given the non-trivial difficulty of the problem—especially because DI is not a metric—we believe it is reasonable and appropriate for performance-engineering improvements to be explored in subsequent work.
> > >
> > > >“why the authors choose to augment HotPotQA on their own”
> > >
> > > For theoretical contributions, it is standard practice to include empirical demonstrations that validate the key claims. Augmenting HotPotQA allowed us to create a redundancy-rich setting in a controlled and reproducible way, and it integrates naturally with our existing codebase. These additional experiments highlight where our method excels (in high-redundancy environments), and complement the results already in the paper showing that our approach remains competitive in system-prompt selection on HotPotQA—even though that setting is inherently query-dependent and therefore less favorable for a query-agnostic method like ours. Together, these results provide a well-rounded picture of the method’s strengths and practical utility.
> > >
> > > Overall, we hope the reviewers will evaluate the work as a **theoretical contribution supported by sufficient empirical evidence** to demonstrate its effectiveness and guide future research. We are sincerely grateful for your thoughtful engagement, and we would be happy to clarify anything further.

---

### Author Response · Authors · 2025-11-24
**To All Reviewers: Additional Experiments**

_Thank you to all reviewers for the constructive feedback. We revamped the experiments with additional redundancy and were able to demonstrate superior performance over PMI._

## Redundancy-Rich Dataset

We appreciate the reviewers’ suggestions to conduct a deeper empirical study on redundancy-rich datasets. To better simulate natural redundancy, we **augmented** HotPotQA as follows: for each sentence in the “supporting facts”, we generated one **paraphrase** and one **summary**. This reflects two realistic forms of redundancy: multiple versions or editions of the same document (captured by paraphrases), and brief mentions of a topic within related contexts (captured by summaries). We used `Llama-3.2-3B-Instruct` to generate both paraphrases and summaries. Examples are available at: https://bit.ly/gamma-cover-augmented-samples (shared anonymously).

We will open-source the augmented dataset after the rebuttal period.

## Clarifying Context Compression and System-Prompt Selection

We also appreciate the feedback pointing out the overlap between **hard compression** and **system-prompt selection** in our initial experimental design. Let $s$ denote the number of supporting facts in the ground truth. We now **redefine** context compression as reducing the number of context chunks to any value $\ge s$, whereas system-prompt selection refers to reducing the number of chunks to $<s$. Note that in HotPotQA, $\bar{s} \approx 2.4$.

We further clarify that we use `Llama-3.2-3B-Instruct` to compute DI and PMI. In later sections, we also provide ablations across different model sizes and model families.

## Results and Findings (with Recall)

We observe across-the-board superior performance over PMI on the augmented dataset, with **up to 10 points of improvement in EM and F1**, and **up to 20 points of improvement in recall**. Note that **recall does not count paraphrases or summaries**. In the table, $k$ denotes the number of context chunks remaining after compression or selection.

$k$ | EM DI | EM PMI | F1 DI | F1 PMI | Recall DI | Recall PMI
-|-|-|-|-|-|-
1 | 33.7$\pm$1.25 | 30.39$\pm$0.85 | 38.42$\pm$0.88 | 35.54$\pm$0.80 | 30.46$\pm$0.29 | 22.38$\pm$0.42
2 | 41.7$\pm$1.00 | 35.04$\pm$0.63 | 46.59$\pm$0.57 | 39.38$\pm$0.29 | 50.21$\pm$0.22 | 37.09$\pm$0.38
$s$ | 44.25$\pm$0.98|36.85$\pm$0.42|48.84$\pm$0.60|40.79$\pm$0.42|55.77$\pm$0.39|40.62$\pm$0.34
$s+1$|50.35$\pm$0.51|41.12$\pm$0.67|54.42$\pm$0.50|44.66$\pm$0.37|69.59$\pm$0.45|51.34$\pm$0.49
$s+2$|53.62$\pm$0.52|47.5$\pm$0.48|57.88$\pm$0.53|50.81$\pm$0.58|78.86$\pm$0.23|64.12$\pm$0.6
$s+3$|55.06$\pm$0.64|50.97$\pm$0.43|59.36$\pm$0.55|54.63$\pm$0.6|85.53$\pm$0.34|74.5$\pm$0.77

## Empirical Study on $\gamma$

We conducted a sensitivity analysis of $\gamma$ using $\gamma \in ${0.5,1,1.5,2}. We observe **an approximately linear relationship between $\gamma$ and both $|S|$ and $\frac{|S|}{|U|}$**, where $S$ is the selected representative set and $U$ is the full set. However, EM, F1, and Recall **degrade superlinearly** as $\gamma$ increases. See Figure 1 at https://bit.ly/gamma-cover-charts (shared anonymously).

$\gamma$ | EM | F1 | Recall | \|S\| | \|S\|/\|U\|
-|-|-|-|-|-
0.5 | 57.61 | 61.38 | 96.46 | 9.398 | 0.821
1|56.34|60.21|91.05|7.751|0.669
1.5|51.50|55.33|75.84|5.384|0.460
2|43.72|47.99|55.26|3.206|0.273

## Empirical Study on $\bar{\delta}$

Estimating $\delta$ is intrinsically difficult because we do not have access to an ideal language model. We restrict attention to paraphrase pairs $(c_i, c_j)$, for which $\gamma \approx 0$ and, for any question $q$, $I(q;c_i) \approx I(q;c_j)$. In this case, the difference $|{\rm PMI}(q;c_i)-{\rm PMI}(q;c_j)|$ is dominated by $\delta$. This estimate is conservative, since in practice we use a non-perfect language model (`Llama-3.2-3B-Instruct`) to generate the paraphrases, which introduces additional noise.

We compute the average value of $\frac{|{\rm PMI}(q;c_i)-{\rm PMI}(q;c_j)|}{|{\rm PMI}(q;c_i)+{\rm PMI}(q;c_j)|}$ over the HotPotQA dataset using three different LLMs. The ratio is consistently below 2%, indicating that $\delta$ is small and leaving ample room for the $|U\setminus S|$ factor.

Model|Avg Ratio
-|-
llama3b|1.75%
llama1b|1.75%
qwen3b|1.79%

## Ablation on Model Sizes and Model Families
We reran the experiments using `Llama-3.2-1B-Instruct` and `Qwen2.5-3B-Instruct` to approximate DI and observed virtually no change in EM or F1, indicating that gamma-covering is **not sensitive to the choice of LLM.**

k|EM Llama3b|Llama1b|Qwen3b|F1 Llama3b|Llama1b|Qwen3b
-|-|-|-|-|-|-
1|33.7|34.2|33.95|38.42|39.35|39.14
2|41.7|42.57|42.11|46.59|47.35|46.8
$s$|44.25|45.24|44.26|48.84|49.68|48.75
$s+1$|50.35|51.09|50.82|54.42|55.22|54.71
$s+2$|53.62|54.72|53.76|57.88|58.62|57.81
$s+3$|55.06|56.05|55.18|59.36|60.06|59.8

---

### Author Response · Authors · 2025-11-29
**To the new AC**

To the new AC

We are very sorry for the situation surrounding the incident. We understand that this situation has created extra work for the ACs, and we are sincerely grateful for your time and care in handling a complex issue.

We are encouraged that the reviewers unanimously recognize the **novelty** of our contribution and its **strong theoretical foundation**. Most of the remaining concerns focus on providing additional empirical evidence demonstrating clear advantages over PMI—a recent SOTA method that also incurs significantly higher online computational cost. In response, we conducted further experiments showing that our approach outperforms PMI in both **context compression** and **system-prompt selection**. We also included ablations demonstrating that our method is **not sensitive to the choice of LLM**, and that the **theoretical bound is not vacuous** in practice.

Taken together, we believe these results provide sufficient and compelling empirical support for **a contribution whose primary value lies in its theoretical framework**. We very much appreciate your willingness to review all materials with fresh eyes and trust your judgment in making a fair and thoughtful recommendation.

Thank you again for the additional effort and for helping uphold the high standards that make ICLR such a strong and principled community.

---

### Meta-Review · Area_Chair_Cogo · 2025-12-30

**Summary:**

The paper presents a novel information-theoretic framework for context selection using Directed Information (DI). The primary motivation is to move context engineering (compression and selection) to an offline, query-agnostic phase, thereby reducing inference-time latency. Reviewers generally praised the theoretical rigor, the novelty of applying DI to this domain, and the clear mathematical grounding (specifically the bridge between DI and PMI).

However, the initial reviews were focused by concerns regarding the narrow empirical evaluation (originally only on the single HotpotQA), the practicality of offline computation in dynamic environments, and a conceptual gap regarding how a query-agnostic method can effectively serve query-dependent tasks. While the authors provided a robust rebuttal with new experiments on an augmented redundancy-rich dataset which is still on the HotpotQA and sensitivity analyses for hyperparameters, the reviewers remained split on whether the empirical evidence sufficiently matched the theoretical ambitions.

**Reviewer Concerns:**

Addressed concerns:
- All reviewers noted that HotpotQA is redundancy-poor. The authors addressed this by augmenting the dataset with LLM-generated paraphrases and summaries. This demonstrated that the proposed method significantly outperforms PMI in high-redundancy settings (improving EM/F1 by ~10 points and recall by ~20 points)
- Reviewers qvkH and eAMx questioned the selection of the tolerance parameter $\gamma$. The authors clarified that a size-based stopping criterion (e.g., "select $k$ chunks") is a practical proxy for $\gamma$ and showed an approximately linear relationship between $\gamma$ and the resulting context size
- Reviewer eAMx was concerned that the theoretical soundness bound might be vacuous due to estimation errors from the LLM. The authors provided an empirical study showing the estimation slack is consistently below 2%, suggesting the theoretical guarantees hold practical weight
- The authors provided an ablation study showing that the performance of $\gamma$-covering is relatively insensitive to the model size used for calculation (1B vs 3B), which helps address cost concerns

Oustanding Concerns:
- Reviewer 6c65 remained fundamentally unconvinced by the use of a query-agnostic method for the query-dependent task of HotpotQA. They argued that without the query signal, the algorithm cannot inherently distinguish Gold Facts from Distractors. While the authors argued that DI-covering prevents redundancy from crowding out diverse facts, this remains a point of philosophical and practical disagreement
- Reviewer yKRm maintained that the cold start problem of recalculating the DI graph when context changes (adding/deleting chunks) is a significant bottleneck. The authors acknowledged this as an engineering challenge for future work, but the reviewer felt it limits the method's current utility in real-world RAG systems
- Despite augmenting HotpotQA, the evaluation still centers on a single base dataset. Reviewers yKRm and qvkH noted that validation across a more diverse range of tasks (e.g., multi-document summarization, long-form QA) would be necessary to fully prove generalizability

**Reviewer Scores:**

Based on the existing discussion between authors and all reviewers, 3 of them already mentioned that they will maintain the original scores after authors' replies with additional experiments. For reviewer yKRm, the redundancy experiments were added but the dynamic overhead concern was not fundamentally solved, which might lead to score unchanged.

---

### Decision · Program_Chairs · 2026-01-26

Reject